# Implementing Big Data Analytics in Marketing Departments: Mixing Organic and Administered Approaches to Increase Data-Driven Decision Making

Devon S. Johnson [1,*], Debika Sihi [2] and Laurent Muzellec [3]

1   Feliciano Business School, Montclair State University, Montclair, NJ 07043, USA
2   Department of Economics and Business, Southwestern University, Georgetown, TX 78626, USA;
    sihid@southwestern.edu
3   Trinity Business School, Trinity College Dublin, D02 PN40 Dublin, Ireland; Laurent.muzellec@tcd.ie
*   Correspondence: johnsonde@mail.montclair.edu

**Abstract:** This study examines the experience of marketing departments to become fully data-driven decision-making organizations. We evaluate an organic approach of departmental sensemaking and an administered approach by which top management increase the influence of analytics skilled employees. Data collection commenced with 15 depth interviews of marketing and analytics professionals in the US and Europe involved in the implementation of big data analytics (BDA) and was followed by a survey data of 298 marketing and analytics middle management professionals at United States based firms. The survey data supports the logic that BDA sensemaking is initiated by top management and is comprised of four primary activities: external knowledge acquisition, improving digitized data quality, big data analytics experimentation and big data analytics information dissemination. Top management drives progress toward data-driven decision-making by facilitating sensemaking and by increasing the influence of BDA skilled employees. This study suggests that while a shift toward enterprise analytics increases the quality of resource available to the marketing department, this approach could stymie the quality of marketing insights gained from BDA. This study presents a model of how to improve the quality of marketing insights and improve data-driven decision-making.

**Keywords:** marketing strategy; big data analytics implementation; data driven marketing

## 1. Introduction

The promise of leveraging big data and analytics capabilities in a firm's strategy has increased the pressure on marketing departments to make data driven analytics central to marketing decision-making. Analytics generally refers to tools that help find hidden patterns in data (c.f. [1], p. 897). Top management has ramped up pressure on marketing departments to develop analytics capabilities that take them from occasional market research studies to continuous use of integrated customer transaction data, including web surfing and social media activity. However, thus far, much of the progress of marketing departments in utilizing big data analytics (BDA) has come primarily from picking the "low hanging fruit" of analytics provided by external digital service suppliers such as advertising and social media platforms. A McKinsey & Company study reports that profiling customers by their web surfing history and customizing digital advertising to reach them can increase digital marketing return on investment (ROI) by 250% [2]. Marketing departments can easily access a variety of external advertising analytics services that optimize targeting and advertising spend among their digital media channels [3].

Industry experts studying BDA implementation have observed that successful implementation is contingent on how well the organization is able to integrate analytics into decision-making [4,5]. Without integration, BDA analytics capabilities become detached

from marketing decision-making and do not achieve a level of routine and ongoing use required to deliver strategic benefits to the organization. Therefore, a robust inhouse analytics capability is necessary. However, the first step for managers is making sense of a rapidly changing BDA phenomenon manifested in a profusion of data sources, new analysis techniques and new software applications. For many organizations, the focal question is how to systematically make sense of a rapidly evolving BDA environment and proceed to become data driven. Prior research (e.g., [6–9]) has identified a variety of factors that impact BDA implementation, including top management, organizational culture, organizational infrastructure and systems characteristics. However, absent from the literature are systematic empirically tested models on how to effectively implement big data analytics especially within the marketing context. How do managers proceed to comprehend the big data analytics environment and pursue implementation? The present article addresses this gap in the literature. Top management may choose to pursue BDA implementation by increasing the influence of BDA skilled employees in marketing decision-making. Alternatively, top management may pursue a more organic approach of engaging the marketing function in a collective effort to make sense of BDA developments. Both approaches are intended to increase the quality of marketing insights and the degree of data driven marketing pursued by firms.

The present research proposes that marketing departments make sense of the evolving BDA analytics environment via key sensemaking behaviors. "Sensemaking is a social process where individuals and groups fashion an understanding of new phenomena through iterative testing of plausible explanations" [10,11]. Secondly, we investigate the effectiveness of BDA skilled employees at generating marketing and engaging in data-driven decision-making within the marketing department of United States based firms. We draw on the information systems (IS) literature in framing the marketing department's approach to understanding BDA as comprehending an emerging organizing vision [12] with the goal of eventual mastery. Organizational commitment to new technology often brings the possibility of multiple outcomes without a clear understanding of the eventual benefits of the technology [10,12]. Interviews conducted with 15 marketing and analytics professionals in the initial stages of this study revealed that many marketing organizations found the pace of change overwhelming and lacked a firm understanding of the resources and expertise involved in becoming a fully data driven organization. Much like the new media revolution before it, BDA implementation will for some time involve extensive discussions and acting on hunches before clarity emerges [13]. While managers may be certain that BDA is essential to future competitiveness, in the initial stages many are uncertain about the techniques and tools that will have sustained relevance and, consequently, where they should place their bets.

Although other theoretical approaches such as technology adaptation [14] absorptive capacity [15] and organizational routines [16] were considered potentially fruitful for explaining BDA implementation, the complexity and rapidly changing nature of the phenomenon made sensemaking theory [17] the most promising. Sensemaking explains how humans engage in actions and speech to retrospectively understand ambiguous circumstances. The aim of the current research is to shed light on the progress of marketing departments in implementing big data analytics capabilities.

This study contributes to research on marketing strategy in multiple ways. First, employees in the marketing function may find it difficult to change from making decisions based on experience to making them based on data, especially when such data is contradictory to the prevailing wisdom [18]. This is made more complicated by the fact that employees who are unskilled in BDA are often unable to visualize the processes involved and the desired outcomes. This research contributes to the existing literature by providing a cogent, empirically tested model for how the marketing department can achieve data-driven decision-making. We articulate the emergence of data-driven decision making as the organizing vision for marketing analytics and identify the flow of activity from initiation to a change in the decision-making culture of the marketing organization.

In so doing, we respond to a call for research to better understand the process by which new metrics and BDA are adopted by marketing departments [19].

The second contribution of this study is that it empirically evaluates two complementary pathways to accomplishing data-driven decision-making within the marketing department. Having realized the importance of BDA to the marketing function, top managers may choose to pursue an *administered* path to implementation involving top management increasing the influence of marketing analytics skilled personnel with the marketing department. Alternatively, top management may pursue an *organic* approach to implementation by involving marketing employees in sensemaking activities to understand and adopt analytics practices leading to a more customized implementation strategy. This is likely to produce an analytics practice that is cognizant of the extant culture of the marketing department. This study demonstrates that while these paths operate in parallel to increased data-driven decision making, they are not equally effective in improving the quality of marketing insights.

The third contribution of this study is to shed light on the influence of BDA employees within the marketing organization. Prior research has examined the contribution of BDA employees to firm effectiveness in broad terms such as the relationship between human capital and data driven management [20]. This is the first study to investigate the influence of BDA employees on the quality of marketing insights. In so doing, we provide early insights on the degree to which marketing departments are making smarter decisions from their analytics investments and not just working harder.

Finally, although experimentation has long been advocated as a rigorous approach to informing marketing decisions [21], its recent popularity in the form of A/B testing resulted from its use by IT professionals to test website design. The product and process adoption literatures have discussed experimentation as an implementation function [22]. Recently, experimentation has become a cost effective "test and learn" capability for modern analytics and marketing departments [23]. Digital environments allow stimuli to be easily administered to experimental and controlled groups with uncontaminated feedback loops to the researcher for tracking response. Hence, the growing importance of experimentation has been recognized. However, research that seeks to understand its role as a systematic integrated capability of the marketing department is lacking. This study contributes to the literature on marketing capabilities by demonstrating the role of experimental orientation as a sensemaking tool within the marketing department. Following a literature review, we discuss organizing vision and sensemaking in providing a theoretical background to our proposed framework of BDA implementation in marketing departments.

## 2. Literature Review

### 2.1. BDA Analytics in Marketing

Research on the effects of emerging BDA on the marketing can be classified into two types, namely conceptual models that provide guidance on how theorists and managers should think about BDA and research that evaluates the effectiveness and value of BDA application [24–27]. Johnson et al. [28] suggest from qualitative research that firms undergo four stages on the way to becoming fully data driven marketers. In the *sprouting stage* firms begin by experimenting with analytical tools. They then proceed to the *recognition stage*, followed by a *commitment stage* and a *culture shift stage* before the fully *data-driven stage*. The fully data driven stage is characterized by use of machine learning, predictive modeling and integration of third-party data among, other things. The authors that progress towards becoming fully data-driven marketers may be impeded by abdication of responsibility for data to external sources, an obsession with ROI and a lack of experimentation.

Research evaluating the effectiveness of BDA in marketing using empirical models and field experiments demonstrate how the nexus of big data and marketing analytics capabilities are applied to increase profits and product development by improving consumer targeting and more effective elicitation of behavioral response (e.g., [29]). Firms with high quality BDA system achieve significant improvements in new product development

quality [25]. The benefits of well applied marketing analytics can be substantial. For example, a study by Jobs et al. [27] looking at big data and advertising analytics concludes that depending on the type of big data firm used, the cost of entry can range from USD 5000 to USD 100,000 a month and deliver mean advertising savings of 20 to 35%. The successful application of BDA in a modern marking environment is contingent on how well firms are able to execute *knowledge fusion* involving the integration of traditional marketing analytics or market research (delayed and sequential data collection) with big data analytics (real time multi-channel data integration and processing) [30]. Firms operating in increasingly complex environments will required a higher degree of knowledge fusion to create value. In the context of advertising, BDA has become a source of monetization of previously less valuable consumer data. It has created targeted dynamic advertising, geo-tracking, and detection of key drivers of consumer behavior from many variables [13]. BDA in advertising has also improved the precision of attributing channel effectiveness and optimizing spend across advertising channels [3]. Iacobucci et al. [31] reminds us that profitability benefits do not come easy. They require substantial customization to product/service context requiring multidisciplinary collaboration even with academic researchers. Johnson et al. [28] suggest that overall success requires that the marketing organization clarifies and balances the relationship between marketing analytics and marketing creativity. The present article is intended to provide managers with key sensemaking factors that aid their understanding of a rapidly changing environment and suggest a path to data driven decision-making.

### 2.2. Theory: Organizing Vision

Early in the diffusion of an IT innovation, a heterogenous network of interests converge around the innovation. This community of interested parties engage in discourse to understand the role and significance of the technology for advancing organizational efficiency. "The organizing vision represents the product of the efforts of members of the community to make sense [32] of the innovation as an organizational opportunity" (c.f. [12], p. 459). According to Swanson and Ramiller [12], in making sense of the innovation the community defines and creates the innovation and thereby determines its future rate of adoption.

We observe that marketing managers have become more resolute regarding the notion that marketing decisions based on data are superior to decisions based on creativity or intuition. Consequently, data-driven marketing has become the organizing vision of marketing BDA. A central objective of marketing departments is achieving competency in BDA. Although, the rationale of data-driven marketing is well understood, considerable ambiguity exists among managers regarding the tools and techniques of BDA and how they inform marketing decision-making. We propose that marketing departments have been engaging in sensemaking activities to improve their understanding of BDA. Swanson and Ramiller [12] propose that the organizing vision facilitates three aspects of IS innovation, namely interpretation, legitimation and mobilization. Organizing visions are formed through interpretation of related events. In the early stages of IS innovation, multiple and seemingly unrelated applications and experimentations often converge to create systematic solutions. The community of interested parties interprets these related activities, creating a logic that becomes the organizing vision. The articulation of this organizing vision surrounding the technology provides the rationale for the widespread adoption of the technology by firms.

An organizing vision is legitimized through its connection to a broader managerial objective. For example, BDA is legitimized by the pursuit of data-driven decision-making in the functional areas of the firm based on the logic that more data reduces over and under investment and improves ROI. Much of the legitimacy for marketing BDA is externally driven with new approaches to advertising and new marketing channel requirements. For example, marketers are abandoning the old swim lane approach to allocating advertising budgets, involving separate tracking of customer response to each promotional channel (e.g., billboard, search, radio advertising) in favor of a BDA approach involving tracking

cross-media effects of advertising and promotions resulting in more precise ROI estimates for each media [3]. Legitimacy of the organizing vision is also achieved through promulgation by those in authority [12]. For instance, C suite executives, as early detectors of the BDA vision, are likely to signal an interest and support resource commitments to departmental BDA sensemaking activities.

Mobilization involves the activation, motivation and structuring of the entrepreneurial community and market forces that support the realization of the objectives of the organizing vision [12]. Blogs, social media, white papers and conferences are the means for rapid diffusion of new trends and terminology. The emergence of open sources programing languages with supporting blogs such as such as R-bloggers (https://www.r-bloggers.com/, accessed on 9 September 2020) and Pythonblogs (http://www.pythonblogs.com/, accessed on 9 September 2020) are the conduits of cross-disciplinary exchange of analytics tools and techniques. These are often the feeding grounds for early interest in analytics and the forerunner to an organized commitment within the marketing department.

*2.3. BDA Sensemaking*

According to enactment theory [7], individuals and departments experiencing equivocal situations and organizational flux (internal and external pressures to change) will engage in sensemaking to reduce ambiguities and to achieve a sense of direction. Their objective is to achieve a sense of rationality and predictability concerning ongoing events. "Viewed as a significant process of organizing, sensemaking unfolds as a sequence in which people . . . engage ongoing circumstances from which they extract cues and make plausible sense retrospectively, while enacting more or less order into those ongoing circumstances" [7], (p. 409). Sensemaking starts by relying on mental models from prior experiences, which are used for *bracketing and labelling* or recognizing, categorizing and naming unfolding events. Sensemaking is retrospective in considering previous observations and interpreting them in light of new understandings. Initial presumptions are tested through experimentation by multiple members of a department to improve the reliability of learning and to enhance information distribution.

Evidence from practitioners suggest that managers are finding BDA implementation to be an equivocal process. BDA is characterized by volume, variety and veracity of data flows, making application of resources intensive and complex [33]. Mimetic pressures [34] emanating from the aggressive pursuit of BDA by competitors and channel partners is forcing managers to implement BDA before the capabilities are fully mastered. A 2014 survey of analytics executives by Ransbotham and colleagues [35] lends support to the unpreparedness of managers by observing that the increasing sophistication of analytics was outpacing the rate at which managers were learning analytics. Consequently, the authors advised that managers need to become comfortable with applying analytical results they do not fully understand. Drawing on enactment theory [11], we argue that marketing departments have been engaging in sensemaking to understand and master marketing BDA analytics.

We observe that marketing departments are performing four primary sensemaking activities to understand and achieve competence in marketing BDA analytics: improving digitized data quality, acquiring external knowledge, disseminating new techniques and experimenting with data. Although these activities have long been identified as capabilities of market-oriented firms [36,37] they have taken on renewed focus under the organizing vision of marketing BDA. Integrated data warehousing and cloud computing has increased data richness, providing a granular 360 degrees view of the customer and his/her lifetime value. These developments have made experimental techniques routine in many marketing departments. In many analytics-oriented departments, it is now accepted that junior staff can challenge their superiors' decisions by providing supporting evidence from A/B tests [38]. Given this context, we propose the conceptual framework and discuss our hypotheses.

## 2.4. Conceptual Framework and Hypotheses

The conceptual framework presented in Figure 1 takes the perspective that top management is the primary initiator of data-driven decision-making within the firm. Top management instructs the marketing department leadership of the need to become more data-driven. The departmental leadership initiates sensemaking to understand the nature and relevance of BDA analytics to the marketing effort. These sensemaking activities can be categorized as external knowledge acquisition, improving digitized data quality, disseminating new techniques and experimenting with data. Simultaneously, top management may increase the influence of BDA skilled employees on making decisions.

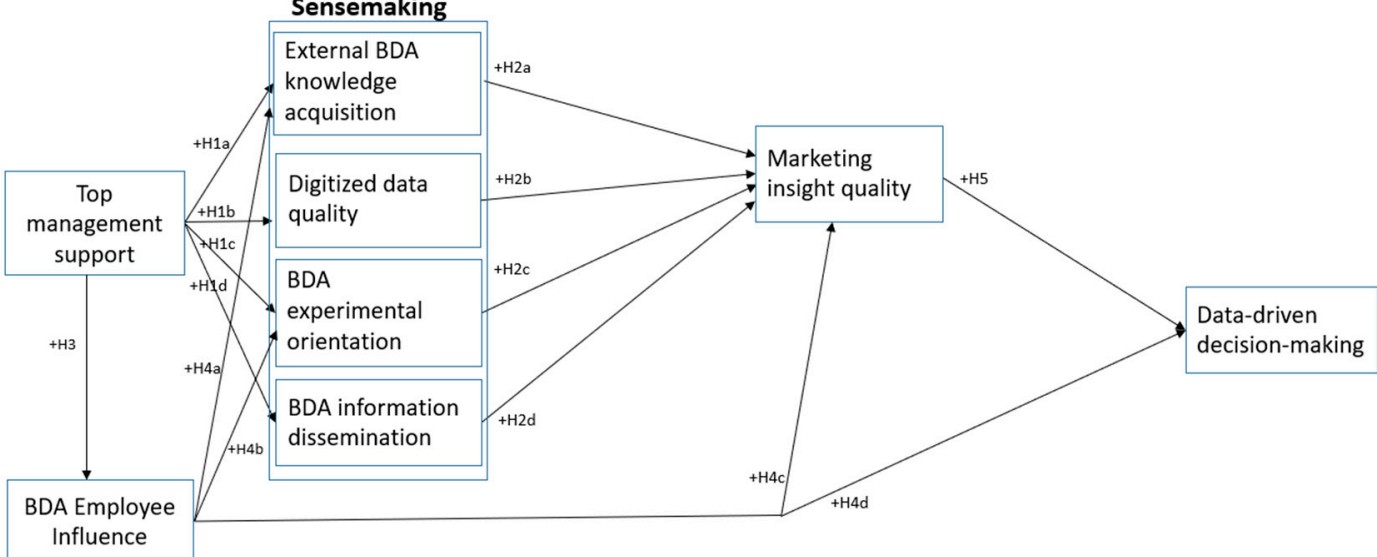

**Figure 1.** Marketing BDA Sensemaking and Data Driven Marketing.

The authors concluded from interviews of professionals involved in marketing related BDA that managers are generally willing to implement BDA analytics. The main challenges confronting them is understanding a rapidly evolving phenomenon and how to effectively move the marketing function toward greater data-driven decision making. Consequently, this study sought to improve our understanding of BDA implementation by testing two implementation approaches. The proposed theoretical framework does not include determinants of willingness to implement technology or implementation success such as user's perceptions, managers attitudes toward new technology [14], adherence to routines made obsolete by new technology [39], psychological safety to engage in experimentation and team stability to facilitate learning [16]. Next, we discuss the relationships hypothesized in Figure 1.

*BDA External knowledge acquisition:* The number of learning sources that firms can draw on for knowledge has been found to be among the most significant determinants of the adoption of data-driven decision-making [20]. External BDA knowledge acquisition refers to the efforts to acquire information on the latest analytical techniques, software applications and sources of data. Among the sources of marketing BDA information are supply chain partners such as advertisers, digital analytics data suppliers, analytics consultants, conference attendance and training programs. The BDA learning process operates in parallel with the ongoing intelligence generation on customers and competitors [37]. The regular market sensing activities of the firm that includes continuous monitoring of customers, competitors, events and trends now shifts attention to understanding competitors' response and intentions toward BDA. As an unfolding area of expertise, marketers are pressured to accelerate departmental learning of BDA principles and the latest "thinking" on how BDA can inform tactical and strategic decisions [40].

*Digitized Data Quality:* When users seek to make sense of IT implementation, they are usually concerned with the quality of the technology and the quality of the information produced by the system [41]. Digitized data quality is access to reliable systems and data for marketing analytics purposes [42,43]. The quality of the information output is an essential determinant of a system's ability to satisfy its customers [42]. Data quality has also been defined as "data that are fit for use by its consumers," implying completeness, accuracy and timeliness of the data [44]. Clean data is required for an effective learning process and the quality of the digitized data is a function of the organization's ability to integrate data from multiple sources. Industry studies have identified data quality and access as key differentiating features of high-performing firms that use BDA to drive innovation [45], Improving data quality is most likely an enterprise initiative requiring the creation of a data warehouse integrating multiple sources of data across departments and from upstream and downstream partners. Therefore, progress on data quality improvement is likely a function of top management's strategic direction and resource commitments.

*BDA Experimental Orientation:* Marketing practitioners have long used experiments as a decision-making aid for multiple reasons. Well-designed experiments allow practitioners to cost-effectively assess product viability with real customers, determine the cause and effect of marketing tactics and reduce risks associated with a full-scale product launch [21]. In many analytics oriented digital firms, experimentation is indispensable to decision-making. These companies have built within their digital infrastructure A/B testing capabilities that allow user involvement in experiments related to user experience, customer engagement and advertising revenue [46]. The innovation literature discusses experimentation as an essential aspect of the implementation process [47,48]. Routinization is achieved through experimentation by adjusting the business processes and organization to reduce ambiguity and improve operational fit [22].

BDA experimental orientation is the commitment of the marketing department to using experiments as a learning vehicle. Experiments are carefully designed, and results are documented for organizational learning. Early in the sensemaking process, as the marketing department learns about big data analytics, they are likely to approach the practice with partially correct or pre-mastery rules [49,50]. This may lead to indecisive decision making. For example, a sales manager may be reticent to ignore insights from a leading salesperson in favor of a next-best-offer prediction by analytic models she may not completely understand.

Experimentation may also benefit the technology implementation process by revealing deficiencies in the social contract of the organization that must be addressed if the innovation is to function effectively. Firms that adopt similar technologies may achieve different levels of performance due to variation in the suitability of the social contracts within the organization [51]. As managers cannot anticipate all situations that may arise, changes in the marketing department's culture are often necessary for innovations to succeed [52]. Designating certain activities as experimental may allow the marketing department to use the implementation process to adjust the organizational culture and improve fit with the innovation. For example, data analytics may show that sending discount coupons to a segment of dissatisfied customers increases subsequent repurchases, whereas calling to apologize produces an insignificant response. However, if the organization is highly customer-centric, not calling a dissatisfied customer may be a violation of the culture. In this case, experiments may be used to demonstrate the effects to employees and to motivate adjustments in organizational culture. Finally, experimentation provides a test-bed for building working relationships between analytics and marketing employees and for enriching cross-departmental relationships. These relational capabilities around data have been agued to contribute to improving decision-making quality [53,54].

*BDA Information Dissemination:* According to organizational learning principles, each employee is a learning agent and over time, the objective is to have all the agents functioning as a shared cognitive system [55]. Organization wide information dissemination provides a shared basis for systematic organizational action [37,56]. The BDA sensemak-

ing process involves integrating information from sources such as conferences, seminars, thought leaders and white papers. Collectively, these sources may not conform to an organizing vision. BDA information flows from external sources to all marketing department employees and insights are generated from application and experiments by each employee. Information dissemination is therefore not restricted to centralized efforts of management but involves bi-directional, vertical and horizontal information flows since each employee is a source of new information and insights. Emerging information may contradict existing policy and culture and may require collective analysis and interpretation to determine its relevance to the marketing context [56,57]. BDA implementation within the marketing department involves group consensus on new analysis techniques, results interpretation and consequential action. As the quality of results by the BDA team improves, new targets and tactical changes must be communicated cross-functionally and vertically to front-line employees. In summary, information dissemination is an essential aspect of the sensemaking process that leads to more insightful decision-making by the marketing department.

*Top Management Support:* Top management is the main initiator of the adoption of new business processes and technologies within organizations in response to external trends or choosing a new strategic direction [14,58]. As boundary spanners with the environment, top managers are often the first to recognize the variety of forces buffeting the organization. When firms are faced with ambiguity and uncertainty involving market opportunities or new technologies, they pursue legitimacy by imitating other successful firms [34]. Top managers sense uncertainties and pursue mimetic change. Top managers impact technology and business process adoption by influencing management training, creating an atmosphere for innovation and by directing resources to support change. Top managers can break down middle management resistance to change and provide relevant technical expertise to facilitate the adoption process [59]. As such, top management support mediates the effects of external forces on the firm's response and determines the pace of assimilation of IT systems such as enterprise resource planning and business processes [60]. These patterns have been observed in the adoption of e-market technologies and online retailing [61].

Top management can initiate focus on BDA techniques by sponsoring senior marketing staff to attend conferences on emerging techniques. Top managers support may also increase the likelihood of marketing departments increasing the use of experiments. This results from the realization that BDA technologies and algorithms reveal the degree to which specific promotion initiatives are responsible for sales increase and how these initiatives may combine to create synergies and therefore improve marketing spend ROI [62]. In summary, we argue that top management support in the form of inspiration and resources is an essential driver of BDA sensemaking within the marketing department. Following from the heretofore discussions the following hypotheses are advanced.

**Hypothesis 1 (H1).** *The higher the level of top management support, the higher the level of marketing BDA sensemaking, specifically (a) external BDA knowledge acquisition, (b) digitized data quality, (c) BDA experimentation orientation and (d) and BDA information dissemination.*

*BDA Insight Quality:* We expect BDA insight quality to be a function of the degree of sensemaking undertaken by the marketing department. Insights are new realizations about the nature of relationships between variables of interest in a decision. Insights provide meaning and significance to data or information [63]. BDA insight quality is the relevance and usefulness of findings from analytics. BDA insights provide new understandings of cause-and-effect relationships among variables and increase understanding of customer behavior and expectations. High quality BDA insights identify the triggers of an intended customer response and when and how to influence behavior most effectively. Increased data flow without more BDA skilled employees and appropriate technology infrastructure will increase confusion and lead to missed opportunities. Sensemaking will increase insight quality by improving data management and analytical skills leading to less biased decisions.

Sensemaking improves the cognitive ability of the marketing department to integrate data from multiple sources. Historically, more data has produced more precise decisions, but this involves a tradeoff whereby the decision maker must apply more cognitive effort [64]. Individuals will therefore balance precision against cognitive effort. It has been argued that marketing data, managerial judgement and supports systems allow firms to diminish this trade-off [65]. A similar reasoning suggests that the organic sensemaking surrounding BDA analytics and the employment of BDA skilled personnel is expected to further diminish the cognitive effort required to accomplish precise decision making. Considering this discussion, we advance the following hypothesis.

**Hypothesis 2a–d (H2a–d).** *The higher the level of marketing BDA sensemaking, specifically (a) external BDA knowledge acquisition, (b) digitized data quality, (c) BDA experimentation orientation and (d) and BDA information dissemination, the higher the level of marketing insight quality to reflect the diagram.*

*BDA Employee Influence:* Finding employees qualified in analytic techniques such as data mining, predictive analytics, data visualization and text analytics is challenging and is likely to remain the case for several years [66,67]. Empirical studies suggest that influence of analytics employees is taking hold across industry. For example, a 2014 MIT Sloan survey of managers at global firms finds that hiring and promoting people with analytics skills and training and integrating new talent into traditional roles are practices associated with the top-performing analytics firms [68]. Brynjolfsson and McElheran [20] find in the manufacturing sector that human capital, defined as the percentage of employees having a bachelor's degree, has a significant positive effect on the extent of data driven decision-making. The premium placed on marketing analytics skills creates the expectation that these employees will have strong influence on decision-making. The intent of top management is that new talent will modernize the marketing function by advocating data-driven over intuition-based decision-making and increase reliance on A/B testing experimental methodologies. BDA employees are a conduit for increasing the flow of knowledge on BDA into the organization to change the culture. The logic is that data analytics skills produce more effective analysis and richer insights for marketing decision-making. This is the motivation for top management to initiate BDA implementation by recruiting analytics skilled employees. As mentioned previously, it is the responsibility of top management to commit resources to creating the capabilities that drive competitive performance [59]. In summary, top management support is expected to directly drive BDA employee influence in areas of external knowledge acquisition and developing and experimental orientation. Additionally, because process implementation within the marketing function is impacted by cultural and structural impediments [69], it is essential to know if the influence of analytics employees is leading to higher quality marketing insights. Thus, we hypothesize the following.

**Hypothesis 3 (H3).** *The higher the level of top management support, the higher the level of BDA employee influence within the marketing organization.*

**Hypothesis 4a–c (H4a–c).** *The higher the level of BDA employee influence, the higher the level of external knowledge acquisition, BDA experimental orientation and marketing insight quality.*

*Data-Driven Marketing:* Accomplishing data-driven marketing is challenging for many firms. At the core of this challenge is the ability to leverage analytics to create business value. LaValle and colleagues [18] observed this to be an obstacle at three stages of analytics adoption that they term *aspirational*, *experienced* and *transformed*. Data-driven marketing requires marketing professionals to complement their traditional creativity and strategy-making skills with new mathematical and software applications that allow them to undertake analysis, interpret findings and initiate strategic and tactical action.

Data-driven marketing is the extent to which strategic and tactical marketing decisions are made based on analysis of customer behavior and cost data. Certain core practices have come to define the pursuit of data-driven marketing. Data-driven marketers have a strong commitment to customer equity and the prioritization of customers according to their lifetime value [69–71]. This philosophical commitment drives the firm to invest in customer relationship management technology. The emergence of activity-based accounting allowed firms to attribute cost to customers more precisely, avoiding overinvestment in low potential customers. Data-driven marketing firms are attentive to selecting customers for acquisition and support in ways that maximize their lifetime value to the firm [72]. Supporting the focus on customer equity is an organizational commitment to using metrics to "keep score" of customer behavior, decision-making and response to marketing initiatives [73]. Marketing metrics is the gathering of data on marketing campaigns, channels, treatments and customer response for customer relationship management [74,75]. Data-driven practitioners are adjusting their loyalty programs from delivering benefits to a customer based on purchase frequency to delivering benefits according to profit realized per customer. The degree of data-driven marketing pursued by a firm is a function of its commitment to market orientation. Research shows that as firms become more customer oriented they display a greater propensity to collect customer data and provide metrics for decision making [76]. Additionally, firms having a strong value delivery culture and robust customer value delivery business processes have been found to use significantly more marketing metrics [75,77]. Market conditions also predict data-driven marketing. Firms that operate in categories requiring high product customization are more highly committed to customer profitability [70].

We argue that during the BDA learning and implementation process, the level of data-driven marketing done by a firm is a function of the quality of the insights generated from its analytics efforts. High quality insights will have a reinforcing effect on the organization's shift to data-driven marketing. We also expect that as BDA employees become more influential within the marketing function, data-driven marketing will increase. Hence, we hypothesize the following.

**Hypothesis 4d (H4d).** *The higher the level of BDA employee influence, the higher the level of data-driven decision-making.*

**Hypothesis 5 (H5).** *The higher the level of BDA insight quality, the higher the level of data-driven decision-making.*

### 3. Methods

We conducted 15 depth interviews of marketing and analytics professionals from the US and Europe involved in the implementation of BDA. The interviews lasted on average one hour. We started the interviews by discussing the meaning of big data analytics within their organization and followed up with questions on championing BDA, BDA organizational learning and BDA progress and challenges encountered in implementation. Responses from the interviews were instrumental in designing the survey.

The survey of key respondents was carried out of 298 marketing professionals drawn from a Qualtrics Inc. panel of business professionals. The survey commenced with a screening question to ensure that data collection was confined to marketing professionals. Specifically, subjects who reported not working in marketing, branding, product management, advertising, social media, market research, marketing analytics, customer relations or sales management were screened out of the study. Table 1 provides details on the industry composition of the sample. Approximately 22% of firms are retail/wholesale, 15% financial business services, 13% technology and telecoms, 10% manufacturing, 9% insurance and 8% marketing advertising. The survey comprised of 53% of firms that are primarily business-to-consumer (B2C) and 47% of primarily business-to-business (B2B) firms. The firms have an average of USD 4 billion in annual sales and a mode of USD 1 billion.

**Table 1.** Sample composition.

| Industries | Percentages |
|---|---|
| Financial/business services | 15% |
| Consumer goods | 5% |
| Healthcare/Pharma | 5% |
| Insurance | 9% |
| Manufacturing | 10% |
| Marketing and advertising | 8% |
| Retail/wholesale | 22% |
| Technology/telecom | 13% |
| Automotive/Transformation | 6% |
| Construction | 4% |
| Other | 9% |

*Measurement*

Top management support was measured using a scale from Karim et al. [78] and demonstrated good composite reliability of 0.91. External knowledge acquisition was measured using three items of Mena and Chabowski's [79] scale to measure stakeholder-focused experiential knowledge acquisition (composite reliability 0.84). The scale measures three ways firms bring knowledge into the organization, namely attending industry events, bringing speakers and experts to the organization and being attentive to industry developments. Digitized data quality was measured using three-items that tapped access to digitized customer data for analytic purposes from Hsieh et al. [42] and Nelson et al. [80]. We added one scale item to capture the availability of integrated digitized customer data. The scale demonstrated high composite reliability of 0.92. BDA dissemination was measured using a 3-item scale (composite reliability 0.90) that drew inspiration from an IT information dissemination scale by Akgün et al. [81]. BDA insight quality is measured using a three-item scale that addresses the quality and usefulness of BDA analysis for marketing decision making (composite reliability 0.91). The scale takes an approach to measuring IT information quality like that used by [80]. BDA employee influence is a three-item scale evaluating the perception that employees with BDA skills have an advantage and are becoming more influential in marketing and business unit decision-making (composite reliability 0.86). The study included three new scales, experimental orientation (composite reliability 0.91), BDA employee influence (composite reliability 0.86) and data-driven marketing (composite reliability 0.87). Construct development involved reviewing the academic and industry literatures to specify the domain of the construct and validating their relevance through interviews with managers [82]. Data-driven marketing was measured using a three-item scale evaluating the degree to which customer facing employees rely on customer data for interactions, serving customers and making suggestions. The nine items of the three scales were subjected to exploratory factor analysis with maximum likelihood estimation and varimax rotation. All variables loaded convincingly on intended constructs with the data-driven marketing factor having item loadings of 0.69, 0.76 and 0.64; experimental orientation with item loadings of 0.79, 0.76 and 0.72. BDA employee influence demonstrated loadings of 0.66, 0.67 and 0.75. Factor cross-loadings ranged from 0.25 to 0.34.

Given that the independent and dependent variables were self-reported using the same survey instrument we investigated the possibility that common-method-variance may have influenced our findings. We assessed the robustness of our results by estimating a single common-method factor in the measurement model and by using the factor scores from this analysis to estimate a structural model that is adjusted for common-method variance [83]. A comparison of the adjusted model with the unadjusted model found no differences in the results. Therefore, we concluded that common method variance does not have a material effect on the findings of our study.

## 4. Results

The mean and standard deviation for each construct as well as the inter-construct correlations are presented in Table 2. We pursued the two-step approach to estimating structural equation models recommended by [26]. Convergent validity was supported by estimating separate measurement models for each construct and ensuring that each measurement item loads on the construct by more than a 0.50 factor loading. All items in the measurement model loaded on intended constructs with factor loadings exceeding the 0.70 threshold (see Table 3). The composite reliability for the constructs range from 0.84 for external knowledge acquisition to 0.92 for digitized data quality and are therefore comfortably above the 0.70 threshold [84]. The average variance extracted (AVE), indicating the percentage of variance attributed to each construct rather than random error, range from 0.63 for external knowledge acquisition to 0.77 for top management support (see Table 3). The AVE for each variable exceeds the square of the correlation between all pairs of constructs in the model, providing support for discriminant validity [85].

**Table 2.** Correlation matrix of variables.

| Variables | Mean | S.D. | 1 | 2 | 3 | 4 | 5 | 6 | 7 | 8 |
|---|---|---|---|---|---|---|---|---|---|---|
| 1. Top mgmt. support | 4.67 | 1.71 | 1 | | | | | | | |
| 2. Ext. know acqu. | 4.74 | 1.46 | 0.52 | 1 | | | | | | |
| 3. Digitized data q. | 4.80 | 1.48 | 0.53 | 0.65 | 1 | | | | | |
| 4. BDA dissemination | 4.78 | 1.50 | 0.47 | 0.68 | 0.70 | 1 | | | | |
| 5. Exp. Orientation | 4.76 | 1.51 | 0.52 | 0.69 | 0.79 | 0.74 | 1 | | | |
| 6. BDA insight | 4.89 | 1.50 | 0.49 | 0.70 | 0.78 | 0.76 | 0.80 | 1 | | |
| 7. BDA employ. infl. | 5.18 | 1.29 | 0.46 | 0.62 | 0.61 | 0.59 | 0.67 | 0.63 | 1 | |
| 8. Data-driven mktg. | 5.01 | 1.40 | 0.38 | 0.59 | 0.61 | 0.62 | 0.67 | 0.68 | 0.68 | 1 |

Notes: Correlations >= 0.09 are significant at $p <= 0.05$.

**Table 3.** Measures.

| | [a] Standardized Loadings | Average Variance Extracted | Composite Reliability |
|---|---|---|---|
| Top Management Support (Karim, Somers and Bhattacherjee, 2007) | | 0.77 | 0.91 |
| TOPM1. Senior executives demonstrate a lot of enthusiasm and interest in marketing related big data analytics implementation | 0.87 | | |
| TOPM2. The overall management support for implementing big data analytics in marketing is quite high | 0.88 | | |
| TOPM3. Upper-level managers have been personally involved in increasing our use of big data analytics. | 0.87 | | |
| External Knowledge Acquisition (Mena and Chabowski 2015) | | 0.63 | 0.84 |
| EKA1. We attend industry events to find out the latest thinking on how big data analytics will be used in the future. | 0.78 | | |
| EKA2. We are attentive to industry changes in big data sources and analytics techniques with potential to influence how we market our products and services | 0.85 | | |
| EKA3. We often bring experts in to speak about the latest developments on marketing/sales related big data and analytical techniques. | 0.75 | | |
| Digitized Data Quality (Hsieh, Rai and Xu 2011; Wixom and Todd 2005) | | 0.74 | 0.92 |

**Table 3.** *Cont.*

| | <sup>a</sup> Standardized Loadings | Average Variance Extracted | Composite Reliability |
|---|---|---|---|
| DAQ1. In terms of system quality, I would rate our access to digitized customer data system highly. | 0.83 | | |
| DAQ2. Regarding the access to digitized customer data, overall its information system is of high quality. | 0.88 | | |
| DAQ3. I would give the quality of customer data we can easily access a high rating. | 0.88 | | |
| DAQ4. We have ready access to integrated digitized customer data for analytics purposes. | 0.87 | | |
| BDA Dissemination (Akgüna et al. 2014) | | 0.74 | 0.90 |
| BDA1. Insights from big data analytics are shared thought the marketing group | 0.85 | | |
| BDA2. New techniques of analysis and reporting are shared throughout the marketing group. | 0.90 | | |
| BDA3. New big data analytics metrics are rapidly deployed across the marketing group. | 0.83 | | |
| Experimental Orientation | | 0.76 | 0.91 |
| EO1 Our marketing analytics staff experiments with new big data analytic techniques to promote learning. | 0.86 | | |
| EO2 Our analytics staff tries various big data analytics techniques to promote learning. (2) | 0.88 | | |
| EO3 Our marketing analytics staff dedicates time and resources to experimenting with big data analytics techniques | 0.88 | | |
| BDA Insight Quality (Nelson, Todd and Wixom 2005) | | | |
| RFQ1. Overall, I would give the quality of insights from our big data analytics efforts a high rating. | 0.85 | 0.76 | 0.91 |
| RFQ2. Overall, our findings from big data analytics efforts have provided very useful insights on how to better market to customers. | 0.90 | | |
| RFQ3. So far, our findings from our big data analytics analysis has had a major impact on our marketing decision-making. | 0.87 | | |
| BDA Employee Influence | | 0.67 | 0.86 |
| EM1. Employees who understand data analytics have a definite advantage in my business unit. | 0.84 | | |
| EM2. Employees who understand data analytics are becoming more influential in my business unit. | 0.80 | | |
| EM3. Influence on marketing decisions is shifting to employees who understand data analytics. | 0.82 | | |
| Data-Driven Marketing | | 0.68 | 0.87 |
| DCDM1. Decisions on how customer facing employees serve customers are dictated by customer data. | 0.81 | | |
| DCDM2. Customer interactions are increasingly driven by the analysis of customer data. | 0.87 | | |
| DCDM3. Customer facing employees have to follow the suggestions predicted by customer data analysis. | 0.79 | | |

<sup>a</sup> All loadings are significant at $p \leq 0.002$. Measurement model: Chi-square = 409 (d.f. = 242, $p$ = 0.000); RMSEA = 0.048; NFI = 0.94; CFI = 0.97; GFI = 0.90.

The hypotheses were tested using structural equations modelling via IBM SPSS Amos 25. First, the measurement model was assessed. All items loaded on their intended construct at a significance level of $p \leq 0.002$. The measurement model demonstrated a $\chi^2$ goodness of fit statistic of 409 with 242 degrees of freedom ($p$ = 0.000), indicating

the measurement model is significant rather having the desired insignificance. However, adequate support is provided by the Root Mean Squared Error of Approximation (RMSEA) having a coefficient of 0.048 indicating an acceptable fit of the data with the measurement model. Additional evidence of acceptable fit of the measurement model with the data is provided by the NFI of 0.94, CFI of 0.97 and the GFI of 0.90 that are equal to or above the 0.90 threshold. Having established the adequacy of the measurement model, we proceeded to estimate a structural model to test our hypotheses.

The structural model demonstrated a $\chi^2$ goodness of fit of 482 (257 d.f., $p < 0.001$). The RMSEA of 0.054 is below the acceptable threshold of 0.08 and the CFI of 0.96, NFI of 0.93 and RFI of 0.92 are above the 0.90 threshold, suggesting a good fit of the model with the data (see Figure 2). Turning to the results, Hypotheses 1a to 1c are confirmed with top management support significantly increasing all sensemaking components specifically, external knowledge acquisition (H1a: $\beta = 0.69$, $p < 0.001$), digitized data quality (H1b: $\beta = 0.90$, $p < 0.001$), experimental orientation (H1c: $\beta = 0.88$, $p < 0.001$) and BDA information dissemination (H1d: $\beta = 0.87$, $p < 0.001$). Regarding hypothesis 2, digitized data quality (H2b: $\beta = 0.25$, $p < 0.001$), experimental orientation (H2c: $\beta = 0.36$, $p < 0.001$) and information dissemination (H2d: $\beta = 0.25$, $p < 0.001$) have significant positive effects on marketing insight quality. However, hypothesis 2a was not supported with external knowledge acquisition not significantly impacting marketing insight quality at the 0.05 level of significance (H2a: $\beta = 0.13$, $p < 0.08$). Hypothesis 3 was confirmed with top management support having a significant positive impact on BDA employee influence (H3: $\beta = 0.77$, $p < 0.001$). Regarding hypothesis 4, BDA employee influence significantly increases the rate of external knowledge acquisition (H4a: $\beta = 0.2$, $p < 0.05$) and data-driven decision-making (H4d: $\beta = 0.5$, $p < 0.001$) but does not significantly affect BDA experimental orientation (H4b: $\beta = 0.08$, $p < 0.29$) and marketing insight quality (H4c: $\beta = 0.00$, $p < 0.89$). Finally, hypothesis 5 was confirmed with marketing insight quality (H5: $\beta = 0.41$, $p < 0.001$) having a significant positive effect on data-driven decision making.

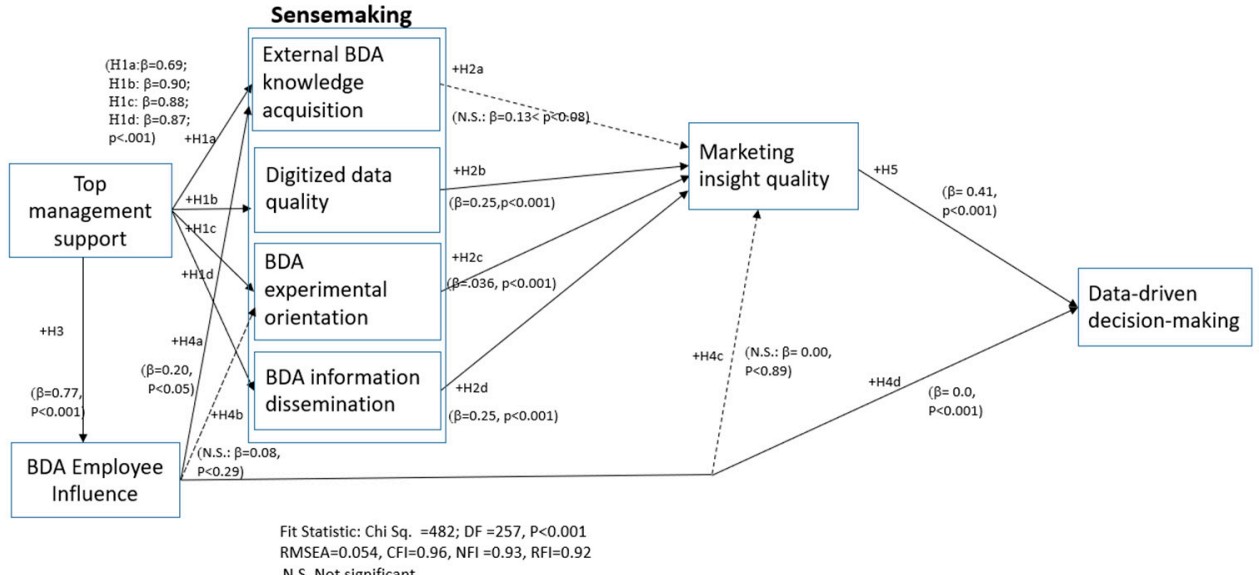

**Figure 2.** Results: Marketing BDA Sensemaking and Data Driven Marketing.

## 5. Discussion

### 5.1. Discussion of Findings

This research examined top management support for BDA analytics sensemaking processes and the influence of analytics employees within the marketing department. It also examined the impact of these factors on data-driven decision-making. Survey data from marketing professionals within the United States provides overall support for the model. First, the results confirmed the effectiveness of taking an organic approach to

achieving data-driven marketing. The results show that top management is a source of motivation for the simultaneous sensemaking activities, namely external knowledge acquisition, digitized data quality, experimental orientation and BDA information dissemination within the marketing department. Top management initiatives such as funding external training programs, conferences and visits by experts are significantly stimulating BDA sensemaking within the marketing department. These findings are consistent with prior studies affirming the role of top management in organizational innovation (e.g., [14,58]. Regarding the consequences of BDA sensemaking, our results show that digitized data quality, BDA experimental orientation and BDA information dissemination lead to significant improvement in the quality of marketing insights, which increases the degree of data driven decision making. External BDA knowledge acquisition has a weak effect on marking insight quality at $p = 0.08$. This weak affect is likely because as external knowledge comes into the organization, it must be adopted to the product context and organizational culture before it can substantially influence performance. Additionally, external knowledge acquisition is often episodic and based on lumpy investments [86] such as acquiring information when external consultants are present or attendance of conferences or training programs. Learning driven by episodic or lumpy expenditures may limit the effect of external information acquisition.

The findings show that the administered approach is only partially effective. Top management support significantly drives BDA employee influence which in turn directly increases the level of data-driven decision-making and external knowledge acquisition. However, BDA employee influence does not directly increase marketing insight quality and experimental orientation. Clearly, the superior approach is for top management to facilitate organic sensemaking processes that increase marketing employee learning and lead to high levels of integration of analytics and domain marketing expertise. This approach will produce a better understanding of when and how BDA may be applied to marketing strategy and create a data culture that has been argued as necessary for effective application BDA [18].

Research suggests that firms engaged in building advanced marketing capabilities are likely to achieve sustained competitive advantage in the later stages from accumulated experience that allows them to climb the learning curve [19]. This implies that the insignificant effect of BDA employee influence on quality of marketing insights is due to BDA employees having insufficient experience within the firm and its business, leading to insufficient accumulated experience. This finding is corroborated by recent industry research that highlights the importance of developing a data culture and the ineffectiveness of building analytics capabilities that are isolated from decision making without important bridging *analytics translator* roles [4]. The data-culture perspective advocates the limited value of data without integration into decision-making and advocates the used of data to support existing decision making, including starting with incremental objectives such as addressing gaps in data availability and speeding up slow processes or removing friction [41]. The goal is early prioritization of data and decision-making integration. The present research lends support to the data-culture perspective.

### 5.2. Managerial Implications

Working as a team, the marketing organization needs to acquire BDA tools, techniques and resource capabilities and figure out how best to apply them to their marketing contexts to generate insights. We suggest that BDA employees minimize separation from the rest of the marketing department and instead promote an organic transformation of the marketing function by engaging and educating their colleagues on analytics principles and by working jointly to determine application cases. Our finding raises doubts about the viability of centralized enterprise analytics before functional areas of the firm have transformed their approach to data-driven decision-making.

We recommend that managers explore ways of incentivizing the marketing department to develop an experimental orientation in BDA. Marketers need to explore and adopt

techniques to meet their demands. Some initiatives may be designated as experimental and assigned lower ROI requirements, allowing flexibility to explore. This may operate similarly to skunk works initiatives like that of Lockheed Martin [87], which allow managers freedom for exploration and experimentation. Procedures for capturing the design and results of each experiment and for making them easily accessible will advance organizational learning. Clear rules on how the results of experiments should be used to inform decision-making should reduce the propensity for middle managers to ignore the results of experiments that contradict established approaches.

We recommend that managers analyze the types of new data and analytical techniques with a view to purposefully understanding their immediate relevance to the sales and marketing efforts. If information acquisition is regarded as only exploratory, early success and direct effects on insight quality may be missed. We recommend that a taskforce be charged with identifying and implementing techniques to improve the capability for marketing insights.

Our results show that the role of top management in making the marketing department more data-driven is mediated by sensemaking processes and by increasing the influence of BDA skilled personnel. Top management is an enabler and can push the process but not pull the marking department toward data driven marketing. We recommend that top management evaluate the analytics capabilities of marketing staff and to address their skill deficiencies. We also recommend that while analytics professionals with deep marketing expertise may be the best fit for the marketing department, marketing professionals with some analytics skills capable of effective interface with enterprise analytics might be a more realistic fit.

### 5.3. Limitations and Directions for Further Research

This study focused on perceptions of BDA implementation within the firm and did not consider the role of specific partner suppliers of analytics. For example, media advertising analytics is most often supplied by external digital agencies that optimize targeting of advertisements. These external partners may increase pressure to adopt analytics. The assessment of analytics supplied by external partners involves sensemaking. We expect that the learning process is accelerated and that decisions are made more quickly. Research is required to understand the influence of external partners on marketing analytics implementation success.

The present study focused on illuminating the implementation process within marketing and as such the dependent variables of interest are quality of marketing insights and level of data-driven decision-making. Ultimately, these outcomes must improve performance in the areas of ROI, margin improvement, customer satisfaction and value delivery. Therefore, we believe there is an opportunity to look at the conditions under which insight quality and data-driven decision-making are likely to be most impactful on firm performance. These conditions could involve factors like environmental turbulence, entrepreneurial leadership and different strategic profiles such as cost leadership, and product or technology leadership.

In this study an administered approach to BDA implementation was defined as top management increasing BDA skilled employees' influence in marketing decision making. In addition to employee influence, measures of BDA skilled employee recruitment would have increased the robustness of the administered approach. Future research should consider this. Finally, our study is the first to systematically examine the growing influence of BDA employees on marketing decision-making. The surprising finding of BDA employee influence having an insignificant effect on marketing insight quality suggest that further investigation is required to better understand how organizational conditions and industry factors impact this relationship. Digital native firms that are built on a software infrastructure with inherent data capabilities clearly have an advantage since analytics is inherent to decision-making in such firms. However, legacy firms transitioning to become

data-driven should be attentive to organic learning processes with functional areas. We hope this work will serve as an impetus for more work in this domain.

## 6. Conclusions

BDA implementation is still unfolding and, as such, interesting opportunities exist for further research. This study examined four aspects of BDA sensemaking taking place within the marketing department. However, because big data requires substantial investments many organizations are shifting to enterprise analytics teams and foregoing departmental level analytics. This study suggests that enterprise analytics may be less organic to departments and therefore achieve quicker transition but only limited improvement in marketing insights. Further research is required to better understand the implications of enterprise analytics for departmental sensemaking and effective data-driven marketing.

**Author Contributions:** Conceptualization, D.S.J. and L.M.; Funding acquisition, D.S. and L.M.; Investigation, D.S.J. and D.S.; Methodology, D.S. and L.M.; Resources, D.S.; Writing—original draft, D.S.J. All authors have read and agreed to the published version of the manuscript.

**Funding:** This research received no external funding.

**Institutional Review Board Statement:** Not applicable.

**Informed Consent Statement:** Not applicable.

**Data Availability Statement:** Data available on request due to restrictions on ethical data management. The data presented in this study are available on request from the corresponding author.

**Conflicts of Interest:** The authors declare no conflict of interest.

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
