# Peer review of "Implementing Big Data Analytics in Marketing Departments: Mixing Organic and Administered Approaches to Increase Data-Driven Decision Making"

_informatics, doi:10.3390/informatics8040066_

Round 1

Reviewer 1 Report

This paper applied structural equation modeling to prove the causal relationship that leads to a data-driven decision. Generally, the paper is well written and formed.
In the causal relationships between construct variables, I agree that top management is the initiator, and they have causal relationships with the different variables that lead to the BDA quality. However, authors may need to revisit the decision of the relationships of construct variables. The weak correlation between top management and employee influence may suggest a different causal relationship. The strong correlation between employee influence and the variables (Ext. know acqu. , etc) may suggest a causal relationship between employee influence and these variables. Further, the direction between the quality insight and employee influence can be the other way round.  
After presenting the results, the authors may need to explain more how this study help the business shift to data-driven decisions. 

Tools used for this research may need to be highlighted (what environment was used for the SEM, R, or other tools?) 

It would be nice if threats to validity were discussed in this research. 

Were there any significant differences detected between different industries? the model may fit some industries more than others. 

Minor issue:
a typo mistake in line 549 (tom management, I guess should be top management)

Author Response

This paper applied structural equation modeling to prove the causal relationship that leads to a data-driven decision. Generally, the paper is well written and formed.
In the causal relationships between construct variables, I agree that top management is the initiator, and they have causal relationships with the different variables that lead to the BDA quality. However, authors may need to revisit the decision of the relationships of construct variables. The weak correlation between top management and employee influence may suggest a different causal relationship. The strong correlation between employee influence and the variables (Ext. know acqu. , etc) may suggest a causal relationship between employee influence and these variables..  

Response: Thank you for your positive and constructive comments. We have been responsive to your comments, and it has improved the article.

Regarding your suggestions on the correlations with employee influences, we have been responsive. We have added 2 hypotheses examining the effect of BDA employee influence on the sensemaking variables of experimental orientation and external information acquisition (H4a and H4b) consistent with your suggestion. We re-estimated the model and report the new results.   

Your suggestion: Further, the direction between the quality insight and employee influence can be the other way round

Response: Quality of insights can affect employee influence as you point out. However, we are studying the implementation process. We think the appropriate modeling is BDA employee influence driving quality of insights during the implementation phase. Top management first must drive change and then after some time transition to an evaluative phase. The relationship will then be reversed (as you suggest) with management asking the question are increased quality insights making BDA employees more influential. We think this is relevant to a post implementation phase.    

After presenting the results, the authors may need to explain more how this study help the business shift to data-driven decisions. 

Response: We have multiple paragraphs in this as follows. In response to your comment, we added another point below that firms should introduce a task force to monitor and improve marketing insight quality. We also tried to make our points clearer. We have also added more literature to provide support and context to our findings. Thank you for this comment.

  1. Discussion

5.1. Discussion of Findings

This research examined top management support for BDA analytics sensemaking processes and the influence of analytics employees within the marketing department. It also examined the impact of these factors on data-driven decision-making. Survey data from marketing professionals within the United States provides overall support for the model. First, the results confirmed the effectiveness of taking an organic approach to achieving data-driven marketing. The results show that top management is a source of motivation for the simultaneous sensemaking activities, namely external knowledge acquisition, digitized data quality, experimental orientation and BDA information dissemination within the marketing department. Top management initiatives such as funding external training programs, conferences and visits by experts are significantly stimulating BDA sensemaking within the marketing department. These findings are consistent with prior studies affirming the role of top management in organizational innovation (e.g. [14, 58]. Regarding the consequences of BDA sensemaking, our results show that digitized data quality, BDA experimental orientation and BDA information dissemination lead to significant improvement in the quality of marketing insights, which increases the degree of data driven decision making. External BDA knowledge acquisition has a weak effect on marking insight quality at p=.08. This weak affect is likely because as external knowledge comes into the organization, it must be adopted to the product context and organizational culture before it can substantially influence performance. Additionally, external knowledge acquisition is often episodic and based on lumpy investments [86] such as acquiring information when external consultants are present or attendance of conferences or training programs. Learning driven by episodic or lumpy expenditures may limit the effect of external information acquisition.

The findings show that the administered approach is only partially effective. Top management support significantly drives BDA employee influence which in turn directly increases the level of data-driven decision-making and external knowledge acquisition. However, BDA employee influence does not directly increase marketing insight quality and experimental orientation. Clearly, the superior approach is for top management to facilitate organic sensemaking processes that increase marketing employee learning and lead to high levels of integration of analytics and domain marketing expertise. This approach will produce a better understanding of when and how BDA may be applied to marketing strategy and create a data culture that has been argued as necessary for effective application BDA [18].

Research suggests that firms engaged in building advanced marketing capabilities are likely to achieve sustained competitive advantage in the later stages from accumulated experience that allows them to climb the learning curve [19]. This implies that the insignificant effect of BDA employee influence on quality of marketing insights is due to BDA employees having insufficient experience within the firm and its business, leading to insufficient accumulated experience. This finding is corroborated by recent industry research that highlight the importance of developing a data culture and the ineffectiveness of building analytics capabilities that are isolated from decision making without important bridging analytics translator roles [4]. The data-culture perspective advocates the limited value of data without integration into decision-making and advocates the used of data to support existing decision making, including starting with incremental objectives such as addressing gaps in data availability and speeding up slow processes or removing friction [87]. The goal is early prioritization of data and decision-making integration. The present research lends support to the data-culture perspective.

5.2. Managerial Implications

Working as a team, the marketing organization needs to acquire BDA tools, techniques and resource capabilities and figure out how best to apply them to their marketing contexts to generate insights. We suggest that BDA employees minimize separation from the rest of the marketing department and instead promote an organic transformation of the marketing function by engaging and educating their colleagues on analytics principles and by working jointly to determine application cases. Our finding raises doubts about the viability of centralized enterprise analytics before functional areas of the firm have transformed their approach to data-driven decision-making. 

We recommend that managers explore ways of incentivizing the marketing department to develop an experimental orientation in BDA. Marketers need to explore and adopt techniques to meet their demands. Some initiatives may be designated as experimental and assigned lower ROI requirements, allowing flexibility to explore. This may operate similarly to skunk works initiatives like that of Lockheed Martin [88] which allow managers freedom for exploration and experimentation. Procedures for capturing the design and results of each experiment and for making them easily accessible will advance organizational learning. Clear rules on how the results of experiments should be used to inform decision-making should reduce the propensity for middle managers to ignore the results of experiments that contradict established approaches.

We recommend that managers analyze the types of new data and analytical techniques with a view to purposefully understanding their immediate relevance to the sales and marketing efforts. If information acquisition is regarded as only exploratory, early success and direct effects on insight quality may be missed. We recommend that a taskforce be charged with identifying and implementing techniques to improve the capability for marketing insights.

Our results show that the role of top management in making the marketing department more data-driven is mediated by sensemaking processes and by increasing the influence of BDA skilled personnel. Top management is an enabler and can push the process but not pull the marking department toward data driven marketing. We recommend that top management evaluate the analytics capabilities of marketing staff and to address their skill deficiencies. We also recommend that while analytics professionals with deep marketing expertise may be the best fit for the marketing department, marketing professionals with some analytics skills capable of effective interface with enterprise analytics might be a more realistic fit.

Tools used for this research may need to be highlighted (what environment was used for the SEM, R, or other tools?) 

Response: Thanks for pointing this out:

Top of page 13: The hypotheses were tested using structural equations modelling via IBM SPSS Amos 25. First, the measurement model was assessed…..

It would be nice if threats to validity were discussed in this research. 

Response: With the structural equations approach the measurement model data is provided on external and internal validity. We also discuss limitations. The results are robust. We also did interviews of managers before to validate the constructs and their contextual relevance.  

Were there any significant differences detected between different industries? the model may fit some industries more than others. 
Response: This is an interesting suggestion. We explored this, but unfortunately, we don’t have enough data on the categories. Our model has a lot of variable and so a small sample size may be problematic.

Minor issue:
a typo mistake in line 549 (tom management, I guess should be top management)

Response: Thank you. Thanks again for your constructive comments

Reviewer 2 Report

This is a data driven research work that carries novelty as it's using real world data. I mainly focused on screening the quality of  presentation and this paper meets my criteria. I would recommend it for publishing. 

Author Response

Thank you very much for taking the time to review our manuscript. We appreciate your support and positive comments. In response to your comments, we have reviewed the manuscript to simplify sentences and make the manuscript more readable. We have also made the positioning/ GAP in the literature clearer and and made extensive changes to the discussion of findings and connections to the literature as requested by other reviewers. Thanks again. 

Reviewer 3 Report

This study aims to explore the processes by which marketing departments make sense of the evolving BDA analytics environment.

My major concerns are expressed below:

1) The topic is, of course, interesting, but it needs to be improved. The authors failed to properly motivate the paper. The paper does not address the "so what?" question. As a result, the implications are modest. The study is based on 298 responses, which is not representative, and as a result, the study's conclusions are not supported by empirical analysis. For me, this is the main limitation of the paper.

2)In the Introduction, you need to connect the state of the art to your paper goals. Please follow the literature review by a clear and concise state of the art analysis. This should clearly show the knowledge gaps identified and link them to your paper goals. Please reason both the novelty and the relevance of your paper goals.

3)The authors just mention that some studies do this and other studies do that but they did not explain the reason for study or study contribution/ Study gap is missing, what the previous studies did not incorporate or how the current is unique from existing literature.

4)The research idea is not properly contextualized, as there is a need of offering a detailed review of relevant literature that help the authors developing the key arguments that support their proposed research.

5) The methodology is at best casual, residing in the use of a mainstream approach. The econometric exercise lacks originality and complexity, but again, the biggest problem is the small number of responses to the Survey.

6) It is important to identify the most important result and explain how the result is obtained and why it matters. Please make sure that your main finding is robust, and you are able to explain the mechanism of the impact with unambiguous supporting evidence in your paper.

7)The conclusions should be further developed in order to explore the main advantages of the approach herein followed. What are the policy implications? What are the main differences regarding the outcomes obtained with this approach and the other approaches available in the scientific literature?

Author Response

This study aims to explore the processes by which marketing departments make sense of the evolving BDA analytics environment.

My major concerns are expressed below:

1) The topic is, of course, interesting, but it needs to be improved. The authors failed to properly motivate the paper. The paper does not address the "so what?" question. As a result, the implications are modest. The study is based on 298 responses, which is not representative, and as a result, the study's conclusions are not supported by empirical analysis. For me, this is the main limitation of the paper.

Response

We respect your opinion, but we don’t agree with it. A study based on 298 responses is not modest and cannot be said to be unrepresentative. We also conducted qualitative research interviewing managers and make sure the issuing our model were the ones managers and marketing organizations are thinking about. The results are robust and well supported by theory. We clearly state and discuss the implications in the contribution in the introduction and in the discussion and managerial implications.

2) In the introduction, you need to connect the state of the art to your paper goals. Please follow the literature review by a clear and concise state of the art analysis. This should clearly show the knowledge gaps identified and link them to your paper goals. Please reason both the novelty and the relevance of your paper goals.

Response

Thank you for specific and constructive feedback. We have clarified the gap in the literature as you suggest. On page 2 we now have the following. 

Industry experts studying BDA implementation have observed that successful implementation is contingent on how well the organization is able to integrate analytics into decision-making [4, 5]. Without integration, BDA analytics capabilities become detached from marketing decision-making and does not achieve a level of routine and ongoing use required to deliver strategic benefits to the organization. Therefore, a robust inhouse analytics capability is necessary. However, the first step for managers is making sense of a rapidly changing BDA phenomenon manifested in a profusion of data sources, new analysis techniques and new software applications. How do firms systematically make sense of the rapidly evolving BDA environment and proceed to become data driven. Prior research (e.g., [6, 7, 8, 9]) has identified a variety of factors that impact BDA implementation (top management, organizational culture, organizational infrastructure, systems characteristics, etc.). However, absent from the literature are systematic empirically tested models on how to effectively implement big data analytics especially within the marketing context. How do managers proceed to comprehend the big data analytics environment and pursue implementation? The present article addresses this gap in the literature. Top management may choose to pursue BDA implementation by increasing the influence of BDA skilled employees in marketing decision-making. Alternatively, top management may pursue a more organic approach of engaging the marketing function in a collective effort to make sense of BDA developments. Both approaches are intended to increase the quality of marketing insights and the degree of data driven marketing pursued by firms.

The present research proposes that marketing departments make sense of the evolving BDA analytics environment via key sensemaking behaviors. “Sensemaking is a social process where individuals and groups fashion an understanding of new phenomena through iterative testing of plausible explanations” [10, 11].  Secondly, we investigate the effectiveness of BDA skilled employees at generating marketing insights and engaging in data-driven decision-making within the marketing department of United States based firms. We draw on the information systems (IS) literature in framing the marketing department’s approach to understanding BDA as comprehending an emerging organizing vision [12] with the goal of eventual mastery. Organizational commitment to new technology often brings the possibility of multiple outcomes without a clear understanding of the eventual benefits of the technology [10, 12]. Interviews conducted with 15 marketing and analytics professionals in the initial stages of this study revealed that many marketing organizations found the pace of change overwhelming and lacked a firm understanding of the resources and expertise involved in becoming a fully data driven organization. Much like the new media revolution before it, BDA implementation will for some time involve extensive discussions and acting on hunches before clarity emerges [13]. While managers may be certain that BDA is essential to future competitiveness, in the initial stages many are uncertain about the techniques and tools that will have sustained relevance and, consequently, where they should place their bets.

Following up on you good recommendation of a state of the art literature review we added the following paragraphs. This covered all the published articles listed on Proquest on the topic of Big Data analytics in Marketing between 2016 and 2021:

                         1.1. BDA Analytics in Marketing 

Research on the effects of emerging BDA on the marketing can be classified into two types, namely conceptual models that provide guidance on how theorists and managers should think about BDA (e.g.[24, 25] Icaobucci et al 2019, Johnson et al) and research that evaluates the effectiveness and value of BDA application [26-29](Ahmad, I. A., Nuseir, M. T., & Alam, M. M. 2021; Xu et al (2016; Jobs; Ahmad, I. A., Nuseir, M. T., & Alam, M. M. (2021) . Johnson et al. [25] suggest from qualitative research that firms undergo four stages on the way to becoming fully data driven marketers. In the sprouting stage firms begin by experimenting with analytical tools. They then proceed to the recognition stage, followed by a commitment stage and a culture shift stage before the fully data-driven stage.  The fully data driven stage is characterized by use of machine learning, predictive modeling and integration of third-party data among, other things. The authors caution that progress towards becoming fully data-driven marketers may be impeded by abdication of responsibility for data to external sources, an obsession with ROI and a lack of experimentation.

Research evaluating the effectiveness of BDA in marketing using empirical models and field experiments demonstrate how the nexus of big data and marketing analytics capabilities are applied to increase profits and product development by improving consumer targeting and more effective elicitation of behavioral response (e.g.[30] Nair ). Firms with high quality BDA system achieve significant improvements in new product development quality [27]. The benefits of well applied marketing analytics can be substantial. For example, a study by Jobs et al.[29] looking at big data and advertising analytics concludes that depending on the type of big data firm used, the cost of entry can range from $5,000 to $100,000 a month and deliver mean advertising savings of 20 to 35%. The successful application of BDA in a modern marking environment is contingent on the how well firms are able to execute “knowledge fusion” involving the integration of traditional marketing analytics or market research involving delayed and sequential data collection with big data analytics involving real time insights from multi-channel integration [31] (Xu et al (2016). Firms operating in increasingly complex environments will required a higher degree of knowledge fusion to create value. In the context of advertising, BDA has become a source of monetization of previously less valuable consumer data. It has created targeted dynamic advertising, geo-tracking, and detection of key drivers of consumer behavior from many variables [13] (Svilar 2013). BDA in advertising has also improved the precision of attributing channel effectiveness and optimizing spend across advertising channels [3] (Nichols 2013). However, Icaobucci et al. [24] cautions that profitability benefits don’t come easy but required substantial customization to product/service context requiring multidisciplinary collaboration even with academic researchers. Johnson et al.[25] suggest that overall success requires that the marketing organization clarifies and balances the relationship between marketing analytics and marketing creativity. The present article is intended to provide managers with key sensemaking factors that aid their understanding of a rapidly changing environment and suggest a path to data driven decision-making.  

3)The authors just mention that some studies do this and other studies do that but they did not explain the reason for study or study contribution/ Study gap is missing, what the previous studies did not incorporate or how the current is unique from existing literature.

Response

The phenomenon of big data analytics implementation is unfolding and will continue to unfold. Is an empirical phenomenon and by its nature a Gap in the literature is implied. It has to be studied repeatedly. This is why we did not emphasize it originally. We have now done so as indicated by our previous response.

4)The research idea is not properly contextualized, as there is a need of offering a detailed review of relevant literature that help the authors developing the key arguments that support their proposed research.

Response: Please see response to 2 above. We also had literature on organizing vision that contextualized the study.

5) The methodology is at best casual, residing in the use of a mainstream approach. The econometric exercise lacks originality and complexity, but again, the biggest problem is the small number of responses to the Survey.

Response: See our response to 1 above and 6 below

6) It is important to identify the most important result and explain how the result is obtained and why it matters. Please make sure that your main finding is robust, and you are able to explain the mechanism of the impact with unambiguous supporting evidence in your paper.

Response:

Findings are explained and tied to literature.

Literature cited in support and explanation of findings:

  1. Fleming, O.F., T.; Henke, N.; Saleh, T. , Ten red flags signaling your analytics program will fail. McKinsey Quarterly, 2018.

  1. Leonard-Barton, D.D., I. , Managerial influence in the implementation of new technology. Management Science, 1988. 34(10): p. 1252-1265.
  2. LaValle, S., Lesser, E., Shockley, R., Hopkins, M. S.; Kruschwitz, N. , Big data, analytics and the path from insights to value. MIT Sloan Management Review, 2011. 52(2): p. 21-32.

  1. Moorman, C.D., G. S. , Organizing for marketing excellence. Journal of Marketing Management, 2016. 80(6): p. 6-38.
  2. Thong, J.Y.L., An integrated model of information systems adoption in small businesses. Journal of Management Information Systems, 1999. 15(4): p. 187-214.

  1. Chahim, M., et al., Product innovation with lumpy investment. Central European Journal of Operations Research, 2017. 25(1): p. 159-182.
  2. Díaz, A.R., K.; Saleh, T. , The Delone and Mclean model of information systems success: A ten-year update. Journal of Management Information Systems, 2018. 19(4): p. 9-30.
  3. Miller, J., Lockheed Martin's Skunk Works: The Official History. 1995: Midland Counties Publications. .

7)The conclusions should be further developed in order to explore the main advantages of the approach herein followed. What are the policy implications? What are the main differences regarding the outcomes obtained with this approach and the other approaches available in the scientific literature?

Response to 6 and 7.

The implications are not for policy makers but for managers. The issue of BDA implementation is corporate/managerial and not government policy.

Reviewer 4 Report

The topic covered in this study is important and interesting.

The methods used by the authors are adequate and allow to obtain some interesting conclusions.

My overall opinion about this study is positive. I have however some concerns that should be addressed before publication:

  • the introduction is too long. In fact, in the current version, it corresponds to around 50% of the text. The authors can create a new section - literature review - using part of the discussion included in the introduction.
  • the discussion of the results could be stronger, namely through a closer link with previous studies on related topics.

Author Response

Comments and Suggestions for Authors

The topic covered in this study is important and interesting.

The methods used by the authors are adequate and allow to obtain some interesting conclusions.

My overall opinion about this study is positive. I have however some concerns that should be addressed before publication:

  • the introduction is too long. In fact, in the current version, it corresponds to around 50% of the text. The authors can create a new section - literature review - using part of the discussion included in the introduction.

Response: Thank you for your positive response. We have tried to follow your recommendation. We have had to add in making the GAP in the literature more pronounced, but we have also added a section in the literature review on BDA in marketing at the request of reviewers. We need to lay out the logic of sensemaking that drives the logic of the article, the model, and hypotheses. Some nuts and bolt explanations are unavoidable.  

  • the discussion of the results could be stronger, namely through a closer link with previous studies on related topics.

Response:

  1. Discussion

5.1. Discussion of Findings

This research examined top management support for BDA analytics sensemaking processes and the influence of analytics employees within the marketing department. It also examined the impact of these factors on data-driven decision-making. Survey data from marketing professionals within the United States provides overall support for the model. First, the results confirmed the effectiveness of taking an organic approach to achieving data-driven marketing. The results show that top management is a source of motivation for the simultaneous sensemaking activities, namely external knowledge acquisition, digitized data quality, experimental orientation and BDA information dissemination within the marketing department. Top management initiatives such as funding external training programs, conferences and visits by experts are significantly stimulating BDA sensemaking within the marketing department. These findings are consistent with prior studies affirming the role of top management in organizational innovation (e.g. [14, 58]. Regarding the consequences of BDA sensemaking, our results show that digitized data quality, BDA experimental orientation and BDA information dissemination lead to significant improvement in the quality of marketing insights, which increases the degree of data driven decision making. External BDA knowledge acquisition has a weak effect on marking insight quality at p=.08. This weak affect is likely because as external knowledge comes into the organization, it must be adopted to the product context and organizational culture before it can substantially influence performance. Additionally, external knowledge acquisition is often episodic and based on lumpy investments [86] such as acquiring information when external consultants are present or attendance of conferences or training programs. Learning driven by episodic or lumpy expenditures may limit the effect of external information acquisition.

The findings show that the administered approach is only partially effective. Top management support significantly drives BDA employee influence which in turn directly increases the level of data-driven decision-making and external knowledge acquisition. However, BDA employee influence does not directly increase marketing insight quality and experimental orientation. Clearly, the superior approach is for top management to facilitate organic sensemaking processes that increase marketing employee learning and lead to high levels of integration of analytics and domain marketing expertise. This approach will produce a better understanding of when and how BDA may be applied to marketing strategy and create a data culture that has been argued as necessary for effective application BDA [18].

Research suggests that firms engaged in building advanced marketing capabilities are likely to achieve sustained competitive advantage in the later stages from accumulated experience that allows them to climb the learning curve [19]. This implies that the insignificant effect of BDA employee influence on quality of marketing insights is due to BDA employees having insufficient experience within the firm and its business, leading to insufficient accumulated experience. This finding is corroborated by recent industry research that highlight the importance of developing a data culture and the ineffectiveness of building analytics capabilities that are isolated from decision making without important bridging analytics translator roles [4]. The data-culture perspective advocates the limited value of data without integration into decision-making and advocates the used of data to support existing decision making, including starting with incremental objectives such as addressing gaps in data availability and speeding up slow processes or removing friction [87]. The goal is early prioritization of data and decision-making integration. The present research lends support to the data-culture perspective.

Literature cited in support and explanation of findings:

  1. Fleming, O.F., T.; Henke, N.; Saleh, T. , Ten red flags signaling your analytics program will fail. McKinsey Quarterly, 2018.

  1. Leonard-Barton, D.D., I. , Managerial influence in the implementation of new technology. Management Science, 1988. 34(10): p. 1252-1265.
  2. LaValle, S., Lesser, E., Shockley, R., Hopkins, M. S.; Kruschwitz, N. , Big data, analytics and the path from insights to value. MIT Sloan Management Review, 2011. 52(2): p. 21-32.

  1. Moorman, C.D., G. S. , Organizing for marketing excellence. Journal of Marketing Management, 2016. 80(6): p. 6-38.
  2. Thong, J.Y.L., An integrated model of information systems adoption in small businesses. Journal of Management Information Systems, 1999. 15(4): p. 187-214.

  1. Chahim, M., et al., Product innovation with lumpy investment. Central European Journal of Operations Research, 2017. 25(1): p. 159-182.
  2. Díaz, A.R., K.; Saleh, T. , The Delone and Mclean model of information systems success: A ten-year update. Journal of Management Information Systems, 2018. 19(4): p. 9-30.
  3. Miller, J., Lockheed Martin's Skunk Works: The Official History. 1995: Midland Counties Publications. .

Round 2

Reviewer 3 Report

Accept in present form.